# Upcycled Animal Feed: Sustainable Solution to Orange Peels Waste

Christina Andrianou, Konstantinos Passadis, Dimitris Malamis, Konstantinos Moustakas, Sofia Mai *
and Elli Maria Barampouti *

Unit of Environmental Science & Technology, School of Chemical Engineering,
National Technical University of Athens, 9 Iroon Polytechniou Str., Zographou Campus, GR-15780 Athens, Greece
* Correspondence: mai@central.ntua.gr (S.M.); belli@central.ntua.gr (E.M.B.); Tel.: +30-21-0772-3243 (E.M.B.)

**Abstract:** Currently, in an effort to increase their sustainability and reduce their carbon footprint, industries look for ways to valorise their waste instead of simply treating it. At the same time, food insecurity is increasing with alarming rates and thus solutions are sought. To this end, the main objective of this paper was to optimise an innovative valorisation strategy to turn orange juice industry by-products into high-value secondary feedstuff for animals. In this context, a valorisation strategy was designed where a saccharification step of the orange peels and an aerobic fermentation step of the liquid residue were included. Both processes were optimised via factorial deign. The saccharification process was optimised in terms of pectinolytic and cellulolytic enzymes and solid loading, whereas the aerobic fermentation method was optimised in terms of nutrients addition, the yeast to glucose ratio, and pH control. According to the optimised conditions, the final animal feedstuff should be formulated by mixing the solid residue of orange peels after the saccharification process under the optimum conditions (50 °C, 24 h, 7.5% solids loading, Pectinex 25 μL/g TS, CellicCTec3 25 μL/g TS), with the harvested yeast cultivated aerobically on orange peels hydrolysate (30 °C, 24 h, orange peels hydrolysate as sugar source, nutrients addition, yeast to glucose ratio equal to 0.02). Finally, the formulated feedstock should be dried in order to stabilise the product in terms of shelf life and feed safety. The final feedstuff presented 23.11% higher in vitro organic matter digestibility and threefold protein content.

**Keywords:** aerobic fermentation; bioeconomy; bioprocess; circular economy; enzymes; feedstuff; sustainable production

## 1. Introduction

Food demand is rising as the world population and thus human food consumption both increase. In the upcoming 30 years, the world's population shall rise by $2 \times 10^9$ people, from $7.7 \times 10^9$ in 2019 to $9.7 \times 10^9$ in 2050, based on the data provided by the United Nations (2019). There is thus a plethora of studies reporting that by 2050 proper measures should be enforced in order to lead to the doubling of world food production needs [1–3]. Hence, as a result of the rising demand, farmers are looking for alternative animal ingredients for the cattle, poultry, and fish nutrition sectors. Based on the predictions of the Food and Agriculture Organization of the United Nations from 2010 to 2050, the demand for food shall rise by 60%. In addition, it is predicted that the demand for animal proteins shall increase by 1.7% per year, with the production of dairy and meat expected to increase by 55% and 70%, respectively [4].

The demand for ruminant feed is primarily driven by the rising per capita meat consumption around the world and the adoption of intensive farming practices. The market for ruminant feed is divided into three categories: animal type (beef cattle, dairy cattle, other), ingredient type (cereals, feed additives, meals, food waste, other), and geography. The feed market of ruminants is poised to report a 3.2% compound annual growth rate.

The worldwide ruminant feed industry is fragmented, with certain feed producers making up a portion of the market while smaller businesses, primarily feed mills, make up the remainder. Feed mills own around 94% of the market share, with the top five firms (Archer Daniels Midland, Chicago, USA; De Heus, Ede, the Netherlands; Land O Lakes Feed, Minnesota, USA; Cargill Inc., Delaware, USA and ForFarmers, Lochem, the Netherlands) holding the remaining 6% [5].

The rising demand for animal-sourced foods has provided favourable circumstances, along with the challenge for producers to increase ruminant feed production in order to render the production sustainable and satisfy the increasing demand of the global dairy and meat sectors globally [4]. Farmers depend on animal feed for their animals to perform well and gain weight in a short period of time. The primary challenge for the farmers is to produce high-quality, healthy meat in order to meet the demand of the meat processing sector [5]. Therefore, this, in turn, may increase the consumption of ruminant feed. The ruminant feed market is expanding given the established practice of on-site mixing of animal feed ingredients by small farmers and livestock producers in order to provide ruminants the nutrients they need in the right amounts. The market for ruminant feed is primarily driven by the rising demand for high-value animal protein, rising consumer awareness of meat and dairy product safety, and the rise in livestock farming [5].

On the other hand, the major restraints are the increasing raw materials cost as well as the legislative constraints. Additionally, consumers across the world are opting for animal-sourced products, such as meat, milk, and other products, that are obtained from organically raised animals. This has resulted in increased sales of organic food products, including dairy and meat products, across the world. This rising demand for organically sourced meat and dairy products may restrain the growth of the compound feed market.

Hence, nutritional manipulation may play a significant role in the profitability and sustainability of the livestock industry. A feasible approach that might lower feed cost while maintaining the nutritional value of the feed is the utilisation of agro-industrial waste as feed ingredients [6–8]. This way, livestock production could meet sustainability via a circular economy approach where the by-products of the food industry could be valorised as secondary feedstuff. However, although currently agro-food industries generate huge amounts of by-products, the later present relatively low interest in the feed industry. It is thus essential to increase their protein content and their digestibility in order to allow for their valorisation.

Orange peels could stand as a possible by-product of the food industry that could be valorised, since they constitute an organic raw material of high value. They are the main by-product of the orange juice industry with high potential as secondary animal feed. From an industrial point of view, an orange can be considered as a composite of 43% juice and 57% peel and pulp. For single-strength juice, 1.33 kg peels are produced per liter of juice, while the respective value for concentrated juice is over 2.85 kg/L. According to the latest FAO statistical data (2019), the geographical distribution of juice production and the respective peels production are presented in Figure 1. The potential of peels production is over 8 million tonnes globally and nearly 600,000 tn in Europe.

Citrus by-products are utilised as a low-cost nutritional supplement to the diets of cattle and have been suggested to inhibit the growth of both Escherichia coli and Salmonella within mixed ruminal microorganism fluid media when supplemented with citrus by-products [9,10]. The incorporation of citrus by-products into diets for cattle have been reported to lead to the reduction of foodborne pathogens due to their antimicrobial properties, since they contain essential oils that possess antimicrobial activities that can damage the cell wall of gram-negative bacteria [9,11]. On the other hand, citrus by-products are also suitable for inclusion in ruminant diets because of the ability of ruminants to ferment high fibre feeds in the rumen [12,13].

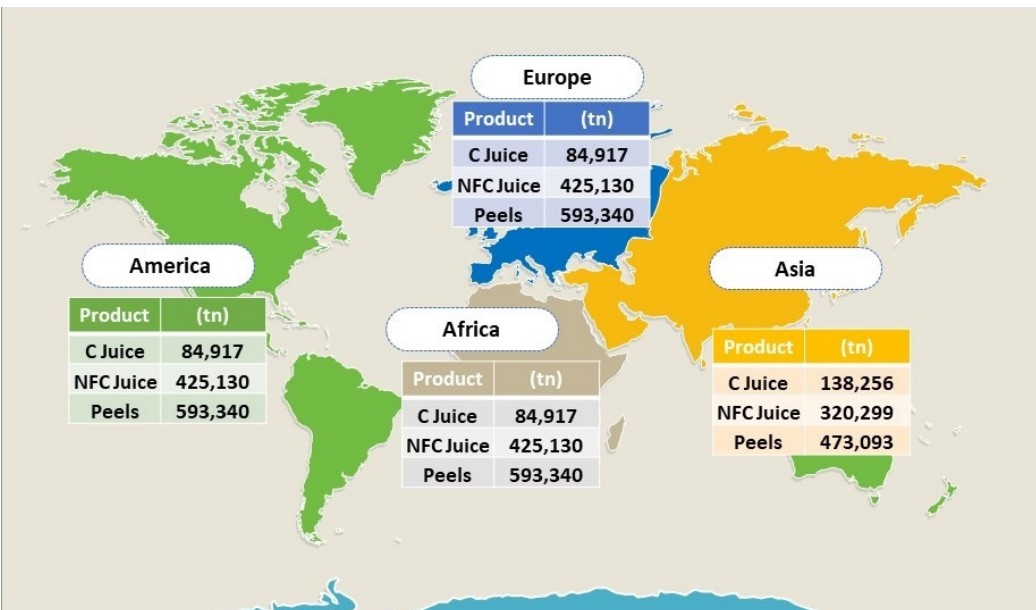

**Figure 1.** Geographical distribution of annual production of orange juice and resulting orange peels (2019).

In view of the above, the main objective of this study was to produce an improved feed ingredient for dairy sheep from orange peels produced by juice industries. In order to receive a nutrient-balanced animal feed ingredient, the pectin and free sugar content of the orange peel waste should decrease and the protein content of the substrate should increase, while preserving or even increasing the digestibility of the feedstock. The optimisation of the process treatment train that included an enzymatic hydrolysis and a fermentation step was performed, resulting in a new secondary feedstuff. Within this study, factorial experimental design was adopted as a methodological approach.

## 2. Materials and Methods

### 2.1. Raw Material

The orange juice industry by-product used in this paper was supplied by Hellenic Fruit Juices Industry, Lakonia, Greece and it was delivered to the Unit of Environmental Science and Technology (UEST), School of Chemical Engineering, National Technical University of Athens.

### 2.2. Physico-Chemical Characterisation

According to the NREL laboratory analytical techniques, moisture, ash, cellulose, hemicellulose, acid-insoluble residue and acid-soluble lignin were assessed [14–17]. A commercial kit (Biosis SA, Athens, Greece) was used to measure glucose using the glucose oxidase-peroxidase technique. The 2019.11 method was used to estimate the amount of ethanol present in the liquid phase [18]. Neutral detergent fibre (NDF), acid detergent fibre (ADF), acid detergent lignin (ADL), crude protein (CP) and crude fat (CF) concentrations were also determined in the samples. Petroleum ether was used to extract crude fat for 6 h, and the Kjeldahl technique was performed to measure nitrogen content (N) [18]. NDF, ADF, and ADL were calculated using techniques from Van Soest et al. [19]. CP was calculated as N $\times$ 6.25. Digestibility is another essential parameter to assess the nutritional value of animal feed. It is related to the energy and nutrients that are accessible to animals [20]. In vitro organic matter digestibility (IVOMD) in the short-term in vitro trial was calculated as described by Pell and Schofield [21].

### 2.3. Experimental Process

In line with the targets described above and after several preliminary trials, a valorisation strategy was designed (Figure 2). Within this strategy, the enzymatic hydrolysis of orange peels was studied, from which a liquid fraction rich in sugars and a hydrolysed solid residue were obtained. The liquid fraction was used for yeast cultivation with the ultimate goal of producing single cell protein. The latter was mixed with the hydrolysed solid residue to produce advanced animal feed. The final feedstuff was dried in order to stabilise the product in terms of shelf life and feed safety. Nutritional and in vitro digestibility value of the final product were set as optimisation parameters.

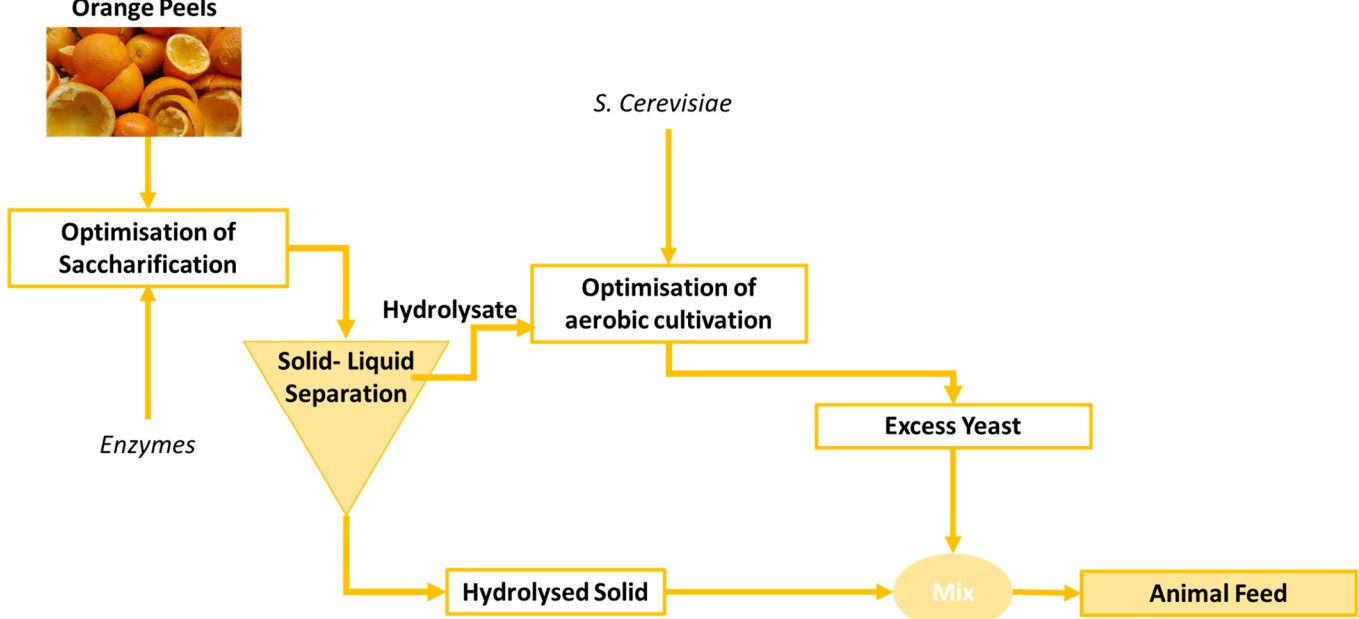

**Figure 2.** Experimental flow chart.

2.3.1. Saccharification Process

A number of 100 mL Duran laboratory borosilicate glass bottles with sealed screw lids are used for the enzymatic saccharification of biomass. The pH of the mixture is measured initially and it is corrected by dilute alkaline solution ($CaCO_3$) in order to set the pH at the optimum range (pH 5.5) of each enzymatic formulation. Cellulose and pectin hydrolysis is performed at 50 °C and 150 rpm for 24 h in a rotary shaker incubator (Constant Temperature Shaking Incubator FS-70B, Ningbo, China) by use of CellicCTec3, Novozymes, Bagsværd, Denmark and Pectinex Ultra, Novozymes, Bagsværd, Denmark. CellicCTec3 is a commercial Novozymes product and is a mixture of cellulases, hemicellulases, and β-glucosidases. A plethora of lignocellulosic feedstocks, pretreated or not, have been shown to respond to its hydrolytic activity, while Pectinex Ultra is a commercial enzymatic formulation that targets the hydrolysis of pectin. For all enzymatic formulations, their activity is estimated based on the standard methods [22]. All experiments are replicated twice, and the mean values are evaluated.

The sugar yield (Ys) is used as a measure of the efficiency of enzymatic hydrolysis which is expressed as the mass of glucose per 100 g of total solids.

A factorial experiment was designed in order to assess the impact of some fundamental process variables on the sugar yield (optimisation parameter). The aim of the factorial experiment at lab-scale was to optimise the dosages of the enzymatic formulations and the solid loading in the saccharification process of orange peels waste, according to Table 1. In the $2^3$ factorial design, 8 experiments were performed in duplicate. Five more experiments were also carried out in the centre of the design for statistical purposes. The exact experiments performed are presented in Table 2. The selection of enzyme loadings

was made based on literature [23] and on preliminary trials (data not shown). A decrease in solid loading was tested given that a further increase would result to inadequate mixing of the mixture. The factorial design was applied as a useful technique to investigate the effect of the process variables on the saccharification process output in terms of sugars yield ($Y_s$). In general, when utilising a $2^n$ factorial design, "n" controlling parameters interact with an optimisation variable via a suitable linear model. Additionally, their importance may be calculated and evaluated [24,25].

**Table 1.** Levels of factorial experiment of orange peels saccharification.

| Parameter | Low Level (−) | High Level (+) | Center (0) |
|---|---|---|---|
| Pectinex (μL/g TS) | 25 | 75 | 50 |
| CellicCTec3 (μL/g TS) | 25 | 75 | 50 |
| Loading (%) | 2.5 | 7.5 | 5 |

**Table 2.** Experiments of factorial experiment of orange peels saccharification.

| Experiments | Pectinex (μL/g TS) | CellicCTec3 (μL/g TS) | Loading (%) |
|---|---|---|---|
| S.1 | 25 | 25 | 2.5 |
| S.2 | 25 | 25 | 7.5 |
| S.3 | 25 | 75 | 7.5 |
| S.4 | 25 | 75 | 2.5 |
| S.5 | 75 | 75 | 2.5 |
| S.6 | 75 | 75 | 7.5 |
| S.7 | 75 | 75 | 7.5 |
| S.8 | 75 | 25 | 2.5 |
| S.9 | 50 | 50 | 5 |

From the results of the factorial experiment, a mathematical model was created and the Fisher criteria was used to assess its adequacy.

2.3.2. Aerobic Fermentation

In 200 mL autoclavable bottles, 100 mL of liquid rich in glucose (from the residue of the enzymatic hydrolysis of orange peels) are added. Nutrients are also added in order to avoid inhibition of the yeast growth according to Salari and Salari [26]. The concentrations of the nutrients added are presented in Table 3.

**Table 3.** Nutrients concentrations added.

| Nutrients | $KH_2PO_4$ | $(NH_4)_2SO_4$ | $MgSO_4$ |
|---|---|---|---|
| Concentration (g/L) | 5 | 2 | 0.4 |

Saccharomyces cerevisiae yeast is then added to all bottles. Finally, air pumps are used for continuous aeration of each sample and the samples are placed in a water bath at 30 °C. After 24 h of aeration, the samples are centrifuged and filtered. The solid residue is placed in pre-weighed capsules and transferred to an oven for 24 h at 50 °C, while the concentration of ethanol and residual glucose is measured in the liquid phase.

Regarding the aerobic fermentation of *S. Cerevisiae*, according to the literature [26–29], pH, DO, and Yeast/Glucose ratio are crucial operational parameters. Moreover, the addition of nutrients is usually beneficial. In order to assess the impact of these parameters on the studied system, a factorial experiment was designed with operational parameters on the addition of

nutrients, the yeast to glucose ratio, and the pH control. In Table 4, the levels of this factorial experiment are presented. The exact experiments performed are presented in Table 5. pH was corrected with $CaCO_3$ to 4.75 when indicated.

**Table 4.** Levels of factorial experiment of aerobic fermentation.

| Parameter | Low Level (−) | High Level (+) |
|---|---|---|
| Nutrients addition | NO | YES |
| Yeast/Glucose (*w/w*) | 0.005 | 0.02 |
| pH control | NO | YES |

**Table 5.** Experiments of factorial experiment of aerobic fermentation.

| No. | Nutrients Addition | Yeast/Glucose (*w/w*) | pH Control |
|---|---|---|---|
| F.1 | YES | 0.020 | YES |
| F.2 | YES | 0.020 | NO |
| F.3 | YES | 0.005 | YES |
| F.4 | YES | 0.005 | NO |
| F.5 | NO | 0.020 | YES |
| F.6 | NO | 0.020 | NO |
| F.7 | NO | 0.005 | YES |
| F.8 | NO | 0.005 | NO |

The levels of the yeast to glucose ratio were predetermined based on preliminary experimentation.

Regarding the optimisation parameter, the aerobic fermentation yield, $Y_{aer}$, is adopted as a measure of biomass production according to the following equation:

$$Y_{aer} = \frac{Biomass\ production}{Total\ Reducing\ Sugars} \left(\frac{g}{g}\right) \tag{1}$$

## 3. Results and Discussion

### 3.1. Orange Peels Composition

The mean values of the main physicochemical and nutritional characteristics of the orange peels waste utilised in this study are illustrated in Table 6.

**Table 6.** Physicochemical and nutritional characteristics of orange peel waste.

| Component | Average | | |
|---|---|---|---|
| TS (%) | 15.5 | ± | 0.8 |
| Moisture (%) | 84.5 | ± | 0.8 |
| Ether extract (% d.b.) | 4.1 | ± | 2.4 |
| Water Soluble Solids (% d.b.) | 25.9 | ± | 7.6 |
| VS (% d.b.) | 95.7 | ± | 0.5 |
| Ash (% d.b.) | 4.3 | ± | 0.5 |
| Cellulose (% d.b.) | 17.2 | ± | 2.7 |
| Hemicellulose (% d.b.) | 37.2 | ± | 3.3 |
| Acid Soluble Lignin (% d.b.) | 1.7 | ± | 0.1 |
| Acid Insoluble Residue (% d.b.) | 15.7 | ± | 6.1 |
| TN (% d.b.) | 1.2 | ± | 0.2 |
| Crude Protein (% d.b.) | 7.2 | ± | 1.0 |
| Neutral Detergent Fibre (NDF) (% d.b.) | 42.8 | ± | 11.2 |
| Acid Detergent Fibre (ADF) (% d.b.) | 30.0 | ± | 5.3 |
| Lignin Acid Detergent (ADL) (% d.b.) | 7.1 | ± | 3.8 |
| In Vitro Organic Matter Digestibility (IVOMD) (% d.b.) | 72.7 | ± | 1.0 |

Cellulose content presents relatively small deviations from the values reported in the literature [30–33] and is within the reported range (13–37% by weight). Regarding lignin, the values found in the literature [30–32] are very close to the experimental values and are within the literature range (2–15% by weight). The same applies for ash. A significant discrepancy with the literature is observed in the value of hemicellulose. This fact could be attributed to different production processes followed in each factory and in general in the different orange varieties. Even the initial in vitro organic matter digestibility of orange peels is high, indicating that even unprocessed orange peels could stand as an interesting ingredient in feedstuff for ruminants. The variety of oranges also plays an important role in the IVOMD as Oloche et al. [34] report when studying different varieties of sweet orange peels. They reported IVOMD values from 63.04 (for mixed sweet orange peels) to 66.71% (Washington orange peels).

### 3.2. Orange Peels Saccharification

At the beginning of each experimental trial, the pH was corrected to 5.5. Nevertheless, by the end of the saccharification process the measured pH was lower (4.1 $\pm$ 0.3), implying the low buffering capacity of the mixture along with the possible production of short chain fatty acids. The hydrolysate in the end of the experimental trials was centrifuged and the liquid and solid phase were characterised.

The following table (Table 7) illustrates the results of the factorial experiment in terms of the concentrations of glucose, total reduced sugars (TRS), and TOC.

**Table 7.** Liquid phase composition of the factorial experimental trials.

| Experiments | Pectinex (µL/g TS) | CellicCTec3 (µL/g TS) | Loading (%) | Glucose (g/L) | | | TRS (g/L) | | | TOC (g/L) | | |
|---|---|---|---|---|---|---|---|---|---|---|---|---|
| S.1 | 25 | 25 | 2.5 | 3.6 | $\pm$ | 0.3 | 6.5 | $\pm$ | 0.1 | 8.3 | $\pm$ | 0.2 |
| S.2 | 25 | 25 | 7.5 | 13.2 | $\pm$ | 0.1 | 20.7 | $\pm$ | 1.8 | 25.9 | $\pm$ | 0.3 |
| S.3 | 25 | 75 | 7.5 | 4.7 | $\pm$ | 0.4 | 6.8 | $\pm$ | 1.0 | 9.0 | $\pm$ | 0.1 |
| S.4 | 25 | 75 | 2.5 | 12.9 | $\pm$ | 0.5 | 20.2 | $\pm$ | 0.9 | 27.9 | $\pm$ | 0.9 |
| S.5 | 75 | 75 | 2.5 | 3.6 | $\pm$ | 0.0 | 4.1 | $\pm$ | 0.7 | 9.5 | $\pm$ | 0.2 |
| S.6 | 75 | 75 | 7.5 | 13.1 | $\pm$ | 0.1 | 21.4 | $\pm$ | 0.2 | 28.8 | $\pm$ | 0.5 |
| S.7 | 75 | 75 | 7.5 | 5.0 | $\pm$ | 0.1 | 7.2 | $\pm$ | 0.6 | 10.0 | $\pm$ | 0.5 |
| S.8 | 75 | 25 | 2.5 | 13.6 | $\pm$ | 0.2 | 28.1 | $\pm$ | 1.7 | 28.4 | $\pm$ | 1.3 |
| S.9 | 50 | 50 | 5 | 9.3 | $\pm$ | 0.0 | 15.0 | $\pm$ | 1.4 | 18.6 | $\pm$ | 0.3 |

From this table (Table 7), it is obvious that the released sugars and overall organic compounds are increased with the solid loading implying that effective solids hydrolysis took place. In all cases, glucose contributed by 19.2 $\pm$ 2.0% to the total organic carbon concentration. This is calculated as the percentage of glucose in terms of carbon that is present in the total concentration of total carbon.

The characteristics of the solid fraction in terms of Total Kjeldahl Nitrogen (TKN), Water Soluble Solids (WS), Cellulose, Hemicellulose, Acid Insoluble Residue (AIR) and Acid Soluble Lignin (ASL) are presented in the following table (Table 8). Given the experimental results and measurements, the degradation efficiencies of total solids, cellulose and hemicellulose along with the sugars yield were calculated and presented in Table 9.

**Table 8.** Solid phase composition of the factorial experimental trials.

| Experiments | Pectinex (μL/g TS) | CellicCTec3 (μL/g TS) | Loading (%) | TN (%) | WS (%) | Cellulose (%) | Hemicellulose (%) | AIR (%) | ASL (%) |
|---|---|---|---|---|---|---|---|---|---|
| S.1 | 25 | 25 | 2.5 | 1.6 ± 0.1 | 14.1 ± 0.6 | 8.9 ± 0.2 | 26.2 ± 3.3 | 23.6 ± 1.2 | 2.1 ± 0.0 |
| S.2 | 25 | 25 | 7.5 | 1.4 ± 0.2 | 29.3 ± 0.3 | 6.4 ± 0.4 | 20.5 ± 0.8 | 20.7 ± 1.4 | 2.0 ± 0.0 |
| S.3 | 25 | 75 | 7.5 | 1.9 ± 0.1 | 11.6 ± 1.1 | 8.9 ± 0.3 | 28.3 ± 0.2 | 23.3 ± 0.1 | 2.3 ± 0.2 |
| S.4 | 25 | 75 | 2.5 | 1.4 ± 0.0 | 28.1 ± 5.7 | 5.8 ± 0.3 | 23.2 ± 4.1 | 19.6 ± 0.2 | 1.6 ± 0.2 |
| S.5 | 75 | 75 | 2.5 | 1.7 ± 0.0 | 17.5 ± 1.1 | 6.5 ± 0.8 | 24.0 ± 1.1 | 25.7 ± 0.5 | 1.9 ± 0.1 |
| S.6 | 75 | 75 | 7.5 | 1.6 ± 0.5 | 30.6 ± 2.9 | 5.6 ± 0.2 | 25.1 ± 2.5 | 20.7 ± 0.8 | 1.6 ± 0.1 |
| S.7 | 75 | 75 | 7.5 | 1.4 ± 0.0 | 12.1 ± 1.5 | 6.6 ± 0.3 | 31.1 ± 3.1 | 22.5 ± 0.3 | 2.2 ± 0.1 |
| S.8 | 75 | 25 | 2.5 | 1.8 ± 0.0 | 30.4 ± 1.8 | 5.2 ± 0.4 | 22.9 ± 0.3 | 19.1 ± 0.5 | 1.7 ± 0.1 |
| S.9 | 50 | 50 | 5 | 1.5 ± 0.0 | 18.8 ± 2.4 | 6.7 ± 0.3 | 28.7 ± 1.6 | 21.4 ± 2.5 | 1.9 ± 0.1 |

**Table 9.** Degradation efficiencies and saccharification yields of the factorial experimental trials.

| Experiments | Pectinex (μL/g TS) | CellicCTec3 (μL/g TS) | Loading (%) | TS Degradation (%) | Cellulose Degradation (%) | Hemicellulose Degradation (%) | Sugars Yield $Y_s$ (g Glucose/100 g TS) |
|---|---|---|---|---|---|---|---|
| S.1 | 25 | 25 | 2.5 | 57.1 ± 0.0 | 75.1 ± 0.6 | 71.0 ± 3.6 | 12.4 ± 1.0 |
| S.2 | 25 | 25 | 7.5 | 42.0 ± 2.9 | 75.8 ± 0.3 | 69.4 ± 0.3 | 14.6 ± 0.2 |
| S.3 | 25 | 75 | 2.5 | 59.4 ± 1.0 | 76.5 ± 1.3 | 70.4 ± 0.9 | 16.6 ± 1.7 |
| S.4 | 25 | 75 | 7.5 | 47.1 ± 0.9 | 80.7 ± 0.5 | 68.4 ± 5.1 | 14.3 ± 0.6 |
| S.5 | 75 | 25 | 2.5 | 59.4 ± 0.7 | 82.9 ± 2.3 | 74.8 ± 1.6 | 12.2 ± 0.1 |
| S.6 | 75 | 25 | 7.5 | 50.4 ± 0.6 | 81.9 ± 0.7 | 68.0 ± 3.6 | 14.4 ± 0.1 |
| S.7 | 75 | 75 | 2.5 | 66.3 ± 1.1 | 79.7 ± 0.6 | 62.3 ± 3.2 | 17.7 ± 0.4 |
| S.8 | 75 | 75 | 7.5 | 52.9 ± 0.8 | 88.7 ± 0.5 | 80.1 ± 0.4 | 15.1 ± 0.2 |
| S.9 | 50 | 50 | 5 | 55.1 ± 1.2 | 81.6 ± 1.4 | 66.7 ± 1.6 | 16.3 ± 0.4 |

By using a certain analytical process [24,25,35,36] and the experimental results, the following mathematical model was developed, interrelating the sugars yield with the controlling parameters of the system:

$$Y_S = 14.46 + 0.95 * X1 - 0.04 * X2 + 0.02 * X3 - 0.1 * X1 * X2 + 0.34 * X1 * X3 - 0.21 * X2 * X3 - 0.59 * X1 * X2 * X3 \qquad (2)$$

The relative size and statistical significance of both the main (b1, b2, b3) and interaction effects (b12, b13, b23, b123) are compared in a Pareto chart of the standardised effects (Figure 3).

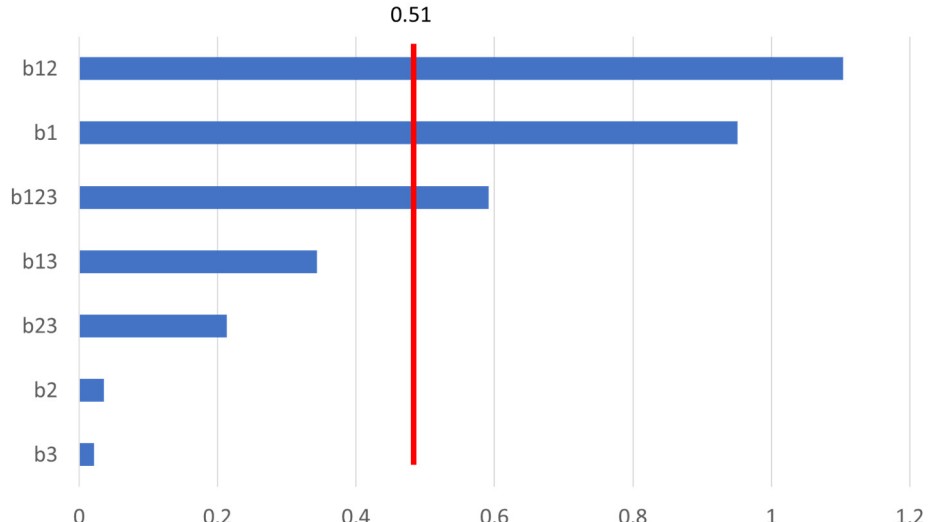

**Figure 3.** Pareto chart of the standardised effects (a = 0.05).

According to these results, the dosage of Pectinex (X1) is statistically significant ($\alpha = 0.05$), along with the interaction of both enzyme dosages and the interaction of all three controlling parameters. In addition, the largest effect poses the interaction of enzymatic formulations (CellicTec3 and Pectinex, both from Novozymes, Bagsværd, Denmark). The effect of the interaction of all parameters (b123) is the smallest because it extends the least.

The plus (+) in the equation above signifies that a decrease in Pectinex dosage would lead to a lower sugars yield and subsequently to lower glucose concentrations. On the other hand, the minus (−) implies a negative impact as, is the case for interactions.

Taking into consideration only the statistical important parameters, the mathematical model is converted into the following equation:

$$Y_s = 14.46 + 0.95X1 - 1.1X1X2 - 0.59X1X2X3 \tag{3}$$

The Fisher criteria was used to determine if the mathematical model created from the factorial design was adequate, and the results showed that it was. Thus, the model could satisfactorily fit to the data within the examined range.

Converting the coded parameters to physical values the mathematical model is:

$$Y_s = 12.88 + 0.0316P - 0.0064C - 0.944L + 0.00013PC + 0.01888PL + 0.01888CL - 0.000378PLC \tag{4}$$

where $Y_s$ saccharification yield expressed as g glucose per 100 g TS.

P Pectinex dosage in μL/g TS

C CellicCTec3 in μL/g TS

L Loading (%).

In the experimental range studied, the highest sugars yield achieved was $17.73 \pm 0.39\%$ in the experimental point 75 μL/g TS Pectinex, 75 μL/g TS CellicCTec3, and 2.5% loading. Nevertheless, the highest glucose concentrations ($13.6 \pm 0.2$ g/L and $13.2 \pm 0.1$ g/L) were observed at S.8 and S.2, respectively, with similarly high sugar yields around 15%. Conclusively, taking into consideration the experimental results and technoeconomic factors (e.g., enzymes cost, minimisation of fresh water needs), the optimum conditions for the saccharification process selected were 50 °C, 7.5% solids loading, Pectinex 25 μL/g TS, and CellicCTec3 25 μL/g TS.

### 3.3. Optimisation of Aerobic Fermentation

The hydrolysate from orange peels at the optimised conditions (50 °C, 7.5% solids loading, Pectinex 25 μL/g TS, CellicCTec3 25 μL/g TS) was tested as substrate for the aerobic cultivation of *S. Cerevisiae*. The results of the factorial experiment of aerobic fermentation are presented in Table 10.

**Table 10.** Results of factorial experiment of anaerobic fermentation of orange peels hydrolysate.

| No. | Nutrients Addition | Yeast/Glucose (*w/w*) | pH Control | Glucose (g/L) | | | Ethanol (g/L) | | | $Y_{aer}$ (g/g) | | |
|-----|-----|-----|-----|-----|-----|-----|-----|-----|-----|-----|-----|-----|
| F.1 | YES | 0.020 | YES | 0.3 | ± | 0.0 | 0.3 | ± | 0.0 | 0.8 | ± | 0.1 |
| F.2 | YES | 0.020 | NO | 0.0 | ± | 0.0 | 0.1 | ± | 0.0 | 0.9 | ± | 0.2 |
| F.3 | YES | 0.005 | YES | 0.0 | ± | 0.0 | 0.0 | ± | 0.0 | 0.6 | ± | 0.1 |
| F.4 | YES | 0.005 | NO | 0.0 | ± | 0.0 | 0.0 | ± | 0.0 | 0.8 | ± | 0.0 |
| F.5 | NO | 0.020 | YES | 0.0 | ± | 0.0 | 0.1 | ± | 0.0 | 0.4 | ± | 0.2 |
| F.6 | NO | 0.020 | NO | 0.0 | ± | 0.0 | 0.1 | ± | 0.0 | 0.2 | ± | 0.0 |
| F.7 | NO | 0.005 | YES | 0.0 | ± | 0.0 | 0.0 | ± | 0.0 | 0.8 | ± | 0.4 |
| F.8 | NO | 0.005 | NO | 0.0 | ± | 0.0 | 0.1 | ± | 0.0 | 0.7 | ± | 0.2 |

From Table 8, it is obvious that in most cases most of the glucose was consumed and minimal ethanol was produced. The only exception was the experiment where all parameters were in the high level (nutrients were added, the yeast to glucose ratio was high, and the pH was controlled). At this experiment, low concentrations of both ethanol and glucose were detected. Nevertheless, relatively high efficiencies were achieved

(0.8 ± 0.0 g/g). The lowest efficiency (0.2 ± 0.0 g/g) was achieved in the experiment where no nutrients were added, the yeast to glucose ratio was high and there was no pH control. However, solid conclusions cannot be drawn from Table 8, but from the mathematical processing of the factorial experiment. Hence, the following mathematical model was developed, interrelating the optimisation parameter with the controlling parameters of the system:

$$Y_{aer} = 0.642 + 0.124 * X1 - 0.07 * X2 + 0.0004 * X3 + 0.178 * X1 * X2 - 0.07 * X1 * X3 + 0.023 * X2 * X3 + 0.002 * X1 * X2 * X3 \quad (5)$$

According to these results, the addition of nutrients (X1) along with its interaction with the yeast to glucose ratio are statistically significant parameters ($\alpha = 0.05$). Most parameters affect the aerobic fermentation yield positively. In addition, the largest positive effect poses the interaction factor.

Taking into consideration only the statistical important parameters, the mathematical model is converted into the following equation:

$$Y_{aer} = 0.642 + 0.124 * X1 + 0.178 * X1 * X2 \quad (6)$$

This model was also proved to be adequate.

In the experimental range studied, the highest aerobic fermentation yield $Y_{aer}$ achieved was 0.9 ± 0.2 g/g in the experimental point where nutrients were added, the yeast to glucose ratio was high (0.02 g/g), and the pH was not controlled. Thus, these conditions were applied in the aerobic fermentation of the orange peels hydrolysate in order to formulate the advanced animal feedstuff.

Prior to the formulation of animal feed, the solid phase that derived after the fermentation of the hydrolysate at the optimum conditions was characterised mainly in terms of total nitrogen and crude protein, in order to assess the potential of its use as a high protein content ingredient. Its nitrogen content was found equal to 7.4% and its crude protein content 46.5%. These characterisation results proved that it is a promising substrate to be mixed with the hydrolysate of orange peels.

According to the results of the experimental trials and the achieved yields, a mass balance of the applied treatment train is presented in Figure 4.

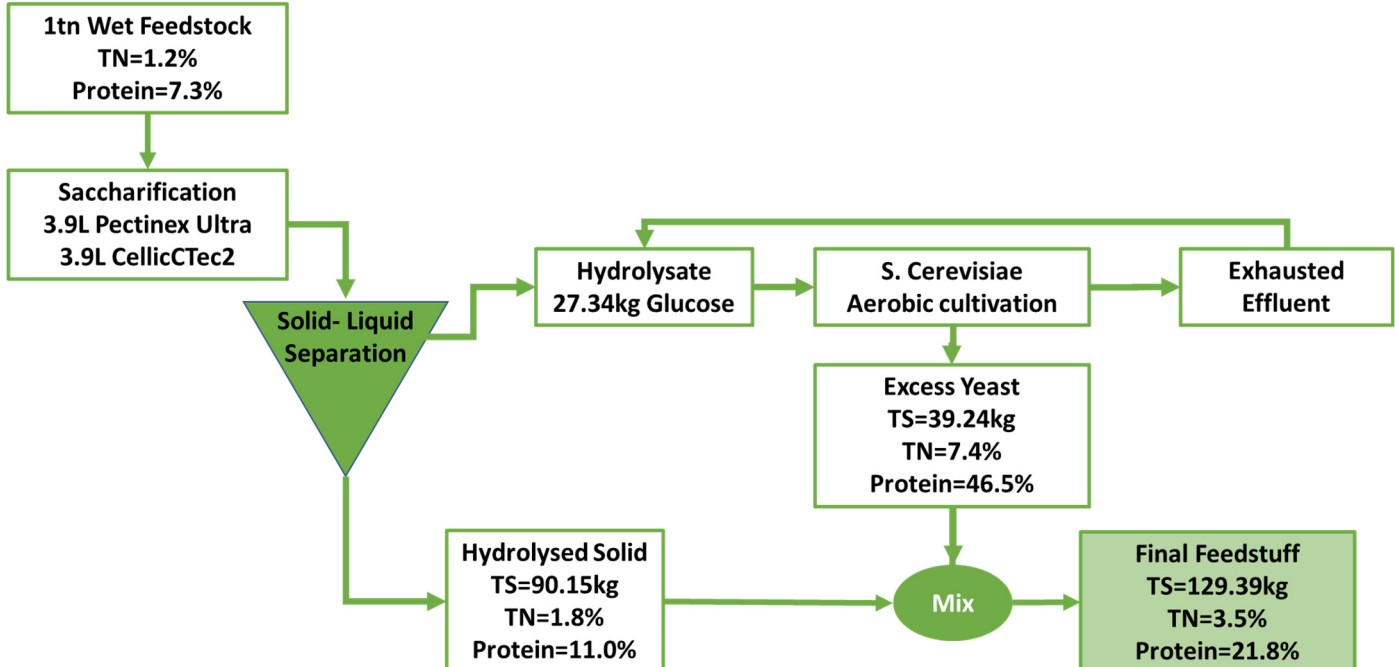

**Figure 4.** Flow chart of strategy applied including mass balances.

It is evident that the implementation of this strategy could lead to a final feedstuff with elevated protein content (21.8%). According to the presented mass balances and in line with a zero waste discharge concept, the mixing of excess yeast and hydrolysed solid (30% yeast) is the final step for the formulation of the animal feedstuff prior to drying.

### 3.4. Formulation of Animal Feedstuff

According to the strategy studied and the results presented above, the final animal feedstuff should be formulated by mixing the solid residue of orange peels after the saccharification process under the optimum conditions (50 °C, 24 h, 7.5% solids loading, Pectinex 25 μL/g TS, CellicCTec3 25 μL/g TS) with the harvested yeast cultivated aerobically on orange peels hydrolysate (30°, 24 h, orange peels hydrolysate as sugar source, aeration, nutrients addition, yeast to glucose ratio equal to 0.02). In this context, a sample of feedstuff prepared under the optimum conditions described above was formulated and characterised in physicochemical and nutritional terms (Table 11).

**Table 11.** Composition of dried unprocessed orange peels and feedstuff prepared under the optimum conditions of the applied strategy.

| Parameter | Feedstuff Prepared under the Optimum Experimental Conditions |
|---|---|
| TS (%) | 94.78 ± 1.22 |
| Moisture (%) | 5.22 ± 1.22 |
| Ash (d.b. %) | 5.03 ± 0.88 |
| VS (d.b. %) | 94.97 ± 0.88 |
| Oil (d.b. %) | 2.25 ± 0.43 |
| TN (d.b. %) | 3.48 ± 0.17 |
| Crude Protein (d.b. %) | 21.77 ± 0.17 |
| Cellulose (d.b. %) | 6.80 ± 0.68 |
| Hemicellulose (d.b. %) | 17.94 ± 1.16 |
| Acid Insoluble Residue (d.b. %) | 18.92 ± 0.56 |
| Ether Extract (d.b. %) | 2.96 ± 0.41 |
| Neutral Detergent Fibre (NDF) (d.b. %) | 28.54 ± 2.31 |
| Acid Detergent Fibre (ADF) (d.b. %) | 17.81 ± 1.98 |
| Lignin Acid Detergent (ADL) (d.b. %) | 6.43 ± 0.17 |
| In Vitro Organic Matter Digestibility (IVOMD) (d.b. %) | 89.5 ± 1.11 |

It is worth noticing that the feedstuff prepared under the optimum conditions of the strategy studied presented higher (23.11%) in vitro organic matter digestibility and almost threefold protein content; thus, the ultimate objective of this work was successfully achieved.

### 4. Concluding Remarks

Conclusively, in this paper an innovative valorisation strategy to turn orange juice industry by-products into high-value secondary feedstuff was presented. Simple biological processes are included that manage to improve the in vitro organic matter digestibility of the initial raw material by almost 25% and to triple the protein content, rendering the orange peel waste an advanced animal feed ingredient.

Nevertheless, the whole value chain of the orange peels consists of six main components and are all essential for adding value and creating a competitive advantage for the new advanced feedstuff. These are the following:

1. Raw material: The raw material producer is mainly the orange juice industry.
2. Receiving, warehousing, maintaining inventory and transportation are all part of inbound logistics.
3. Operations include all the processes necessary to convert orange peels into products—animal feedstuff.
4. Activities such as the distribution of the finished products to a customer are included in outbound logistics.

5. Strategies used in marketing and sales to increase visibility and target the right consumers include price, promotions, and advertising.
6. The end-users, which are mainly animal feed companies, animal feed distributors, farmers and sheep products consumers.

Thus, it is not enough to find a suitable solution regarding one component along the value chain; each step along the value chain must be examined and its weaknesses must be identified and addressed. However, when the core issues, such as the operations step, have been properly tackled, sustainable solutions may thrive.

**Author Contributions:** Conceptualization, S.M. and E.M.B.; methodology, K.P.; formal analysis, K.P.; investigation, C.A. and K.P.; resources, D.M. and K.M.; data curation, K.P., S.M. and E.M.B.; writing—original draft preparation, S.M. and E.M.B.; writing—review and editing, S.M. and E.M.B.; visualization, S.M. and E.M.B.; supervision, S.M. and E.M.B.; project administration, D.M. and K.M.; funding acquisition, D.M. and K.M. All authors have read and agreed to the published version of the manuscript.

**Funding:** The research leading to these results has received funding from the European Union's PRIMA Program for Research, Technological Development and Demonstration under grant agreement n°2013 (NEWFEED, https://newfeed-prima.eu/).

**Institutional Review Board Statement:** Not applicable.

**Informed Consent Statement:** Not applicable.

**Data Availability Statement:** Not applicable.

**Conflicts of Interest:** The authors declare no conflict of interest.

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
