# Peer review of "Upcycled Animal Feed: Sustainable Solution to Orange Peels Waste"

_sustainability, doi:10.3390/su15032033_

Round 1

Reviewer 1 Report

The researchers conducted a research on "Upcycled animal feed: Sustainable solution to orange peels waste"However it lacks major experiments which are important for animals feed, i suggest you to revise the manuscript as per suggestions and add the suggested results accordingly;

Please revise the abstract section add results and MM, it lack proper information revise it please 

keywords must be changed 

Introduction is poor written please revise and brief it also add some information which is related importance of the orange peel waste in animal diet, add literature and ref. 

2.2 Physico-chemical characterization 

please add the brief procedure of each test, do not mentioned only as per........ revise 

what about the invitro rumen fermentation i suggest please add the invitro rumen fermentation, because this is very important if you are providing the orange peel waste. add the materials methods section also results.

Results sections must be revised as per suggestion and without the invitro fermentation you can not recommonded   this by product in animal feed, the discussion section is poor written must be revised and also revised the conclusion section, revised the manuscript add the important parameters like in-vitro rumen fermentation.

please prepare the manuscript as per journal format 

thanks 

Author Response

Reviewer 1

The researchers conducted a research on "Upcycled animal feed: Sustainable solution to orange peels waste". However it lacks major experiments which are important for animals feed, i suggest you to revise the manuscript as per suggestions and add the suggested results accordingly;

Please revise the abstract section add results and MM, it lack proper information revise it please 

The abstract and materials and methods sections were revised according to the reviewer’s comment.

keywords must be changed 

Some keywords were changed as suggested.

Introduction is poor written please revise and brief it also add some information which is related importance of the orange peel waste in animal diet, add literature and ref. 

The introduction was revised and information was added on the orange peel waste in animal diet.

2.2 Physico-chemical characterization 

please add the brief procedure of each test, do not mentioned only as per........ revise 

The methods used for the characterization of feedstocks, intermediate products and end-products are standard methods and thus to the authors’ point of view do not need further elaboration. If however, the reviewer believes that further information should be added, this is not an issue.

what about the invitro rumen fermentation i suggest please add the invitro rumen fermentation, because this is very important if you are providing the orange peel waste. add the materials methods section also results.

The in vitro organic matter digestibility is a similar parameter that is used to report the impact of the processed orange peels on the animals.

Results sections must be revised as per suggestion and without the invitro fermentation you can not recommonded   this by product in animal feed, the discussion section is poor written must be revised and also revised the conclusion section, revised the manuscript add the important parameters like in-vitro rumen fermentation.

The results and conclusions sections were revised according to the reviewer’s comment.

please prepare the manuscript as per journal format 

The manuscript is now prepared following fully the journal format.

thanks 

Reviewer 2 Report

In the manuscript entitled, Upcycled animal feed: Sustainable solution orange peel waste, the authors describe the process of optimizing and valorization of an innovative strategy for converting waste products from the orange juice industry into high-quality animal feed.

The valorization strategy was carried out by pursuing an increase in the sugar content (saccharification) in the orange peel waste and subsequent aerobic fermentation of the resulting liquid residue.

To this end, they conducted a factorial experiment in which they monitored the process of saccharification by the hydrolysis of cellulose and pectin with two types of enzymes. The content of these enzymes (Pectinex and CellicCTec3) and the amount of orange peel treated were the factors. Based on the obtained results they conclude that it's possible to produce feed containing up to 25% more digestible protein from orange peel waste.

Comments:

The introduction, the aim of the work, and the description of the methods are generally clearly presented. Also, the presentation of the results is traceable but discussions seem to be overemphasized and overloaded with numerical data which makes the conclusions blurred. Please refer to pages 9, 12, and 13, and, the concluding remarks.  

1.       The main objection relates to the presentation of the results obtained by the experimental techniques used and the factorial analysis, which are presented in Tables 4-8 and, through the manuscript form. It refers in particular to the number of decimal places you give. You give values with two or three decimal places for the measured parameters as well as for the measurement errors (standard deviation). The number of significant decimal places given is excessive. For most measurement procedures in analytics, one assumes an error of 3-5%. Please correct accordingly.

2.       Page 7.  Your statement: From this Table (Table 5), it is obvious that the released sugars and overall organic compounds are increased solid loading implying that effective solids hydrolysis took place. In all cases, glucose contributed by 19.19±2.00% to the total organic carbon concentration.   Your statement: From this Table (Table 5), it is obvious that the released sugars and overall organic compounds are increased solid loading implying that effective solids hydrolysis took place. In all cases, glucose contributed by 19.19±2.00% to the total organic carbon concentration.   From Table 5 I cannot recognize the value shown. Clarify, please. Again, related to comment 1, if the error in concentration is 2% the value should be expressed as 19%.

3.       Page 9. You start with: By using a certain analytical process 19,20 , please describe in more detail (a certain) analytical procedures used. The references given are dating to 1975 and 1957 so I suppose you used a kind of software available you should refer to.

4.       Page 9. Standardize the syntax. In all the equations you marked the factors as X1, X2, and X3, while in Figure  2 you use the marks b1, b2, b3, b12, ...

5.       Page 9. You say: Converting the coded parameters to physical values… How did you do the conversion? Do explain.

Author Response

Reviewer 2

In the manuscript entitled, Upcycled animal feed: Sustainable solution orange peel waste, the authors describe the process of optimizing and valorization of an innovative strategy for converting waste products from the orange juice industry into high-quality animal feed.

The valorization strategy was carried out by pursuing an increase in the sugar content (saccharification) in the orange peel waste and subsequent aerobic fermentation of the resulting liquid residue.

To this end, they conducted a factorial experiment in which they monitored the process of saccharification by the hydrolysis of cellulose and pectin with two types of enzymes. The content of these enzymes (Pectinex and CellicCTec3) and the amount of orange peel treated were the factors. Based on the obtained results they conclude that it's possible to produce feed containing up to 25% more digestible protein from orange peel waste.

Comments:

The introduction, the aim of the work, and the description of the methods are generally clearly presented. Also, the presentation of the results is traceable but discussions seem to be overemphasized and overloaded with numerical data which makes the conclusions blurred. Please refer to pages 9, 12, and 13, and, the concluding remarks.  

  1. The main objection relates to the presentation of the results obtained by the experimental techniques used and the factorial analysis, which are presented in Tables 4-8 and, through the manuscript form. It refers in particular to the number of decimal places you give. You give values with two or three decimal places for the measured parameters as well as for the measurement errors (standard deviation). The number of significant decimal places given is excessive. For most measurement procedures in analytics, one assumes an error of 3-5%. Please correct accordingly.

The significant decimal places in Tables 4-8 were corrected according to reviewer’s comment. The standard deviations/errors are not presented as percentages of the values but as concentrations or contents (the same as the measured parameters).

  1. Page 7.  Your statement: From this Table (Table 5), it is obvious that the released sugars and overall organic compounds are increased solid loading implying that effective solids hydrolysis took place. In all cases, glucose contributed by 19.19±2.00% to the total organic carbon concentration.   Your statement: From this Table (Table 5), it is obvious that the released sugars and overall organic compounds are increased solid loading implying that effective solids hydrolysis took place. In all cases, glucose contributed by 19.19±2.00% to the total organic carbon concentration.   From Table 5 I cannot recognize the value shown. Clarify, please. Again, related to comment 1, if the error in concentration is 2% the value should be expressed as 19%.

You are correct. 19.19% is not presented in Table 5. It is calculated as the percentage of glucose in terms of carbon that is present in the total concentration of total carbon. This is now also explained in the manuscript.

Regarding the error, it is not 2% of 19.19 but the value ranges from 17.19 to 21.19%.

  1. Page 9. You start with: By using a certain analytical process 19,20, please describe in more detail (a certain) analytical procedures used. The references given are dating to 1975 and 1957 so I suppose you used a kind of software available you should refer to.

These two citations refer to the standard processing of a factorial experiment. Two more recent papers of our own are also inc..luded in the reference list where more information can be found on the processing of the results from a factorial experiment. A dedicated software such SPSS was not used, just simple mathematical tools like Microsoft Excel. To the authors’ point of view, there is no need to include this information in the manuscript. If the reviewer believes otherwise, of course we have no objections on this.

  1. Page 9. Standardize the syntax. In all the equations you marked the factors as X1, X2, and X3, while in Figure  2 you use the marks b1, b2, b3, b12, ...

      X1, X2 and X3 are the parameters of the factorial experiment (Pectinex, CellicCTec3, Loading) whereas b1,b2 and b3 are the main and b12, b13, b23, b123 the interaction effects. This is now more clearly explained in the manuscript.

  1. Page 9. You say: Converting the coded parameters to physical values… How did you do the conversion? Do explain.

 This is part of the factorial experiment methodology. More information can be found in the citations reported.

Reviewer 3 Report

This article describes a way of putting orange peel to efficient use. In this paper, the saccharification process of orange peel and the aerobic fermentation steps of the liquid residue are optimized and the saccharified solid residue is made into a nutrient-rich animal feed. However the following issues should be addressed before it could be published.

Detailed Comments:

1. In the third paragraph on page 2, there is a growing consumer demand for organic food, but there is no explanation of how orange peel feed relates to organic food. Please explain this concept in the appropriate place.

2. It is well known that the cellulose, hemicellulose and lignin content varies in different species of plants. In the section of 2.2, what is the significance of proposing an evaluation of moisture, ash, cellulose, hemicellulose, acid insoluble residues and acid soluble lignin in orange peel?

3. Would it be possible to add an experimental flow chart to the 'Materials and Methods' section to describe the experimental process?

Author Response

Reviewer 3

This article describes a way of putting orange peel to efficient use. In this paper, the saccharification process of orange peel and the aerobic fermentation steps of the liquid residue are optimized and the saccharified solid residue is made into a nutrient-rich animal feed. However the following issues should be addressed before it could be published.

Detailed Comments:

  1. In the third paragraph on page 2, there is a growing consumer demand for organic food, but there is no explanation of how orange peel feed relates to organic food. Please explain this concept in the appropriate place.

Some information on how orange peel feed relates to organic food was added in the introduction.

  1. It is well known that the cellulose, hemicellulose and lignin content varies in different species of plants. In the section of 2.2, what is the significance of proposing an evaluation of moisture, ash, cellulose, hemicellulose, acid insoluble residues and acid soluble lignin in orange peel?

In order to perform the saccharification of orange peels in lab scale, it is necessary to know the exact composition of the substrate. If such a strategy is applied in large scale, it will not be necessary to perform an analytical characterization each time. But to the authors’ opinion, it is necessary in a research paper to include such a Table. But if the reviewer insists, we can omit it.

  1. Would it be possible to add an experimental flow chart to the 'Materials and Methods' section to describe the experimental process?

Figure 2 (Experimental flow chart) was added in the Materials and Methods section according to the reviewer’s recommendations.

Round 2

Reviewer 1 Report

The authors have revised the manuscript but many points are not incorporated so please incorporate all the suggestions 

In previous comments i suuggested you to  add the material method few lines in abstract please add according to previous suggestion.

Objectives must be clear and should be revised please 

it is better and mandatory to add brief procedure of each test which you have performed in materials method section.

what about the invitro rumen fermentation i suggest please add the invitro rumen fermentation, because this is very important if you are providing the orange peel waste. add the materials methods section also results.

response: The in vitro organic matter digestibility is a similar parameter that is used to report the impact of the processed orange peels on the animals.

But i did not get the results which are related fermentation please revise accordingly.

Change Animal figure add any other clear picture in figure 2.

please mentioned all tables and figures in  text correctly.
